# Antioxidant Phytochemicals as Potential Therapy for Diabetic Complications

**DOI:** 10.3390/antiox12010123

**Published:** 2023-01-04

**Authors:** Oke-Oghene Philomena Akpoveso, Emeka Emmanuel Ubah, Gideon Obasanmi

**Affiliations:** 1School of Basic Sciences, All American Institute of Medical Sciences, Black River, Saint Elizabeth Parish JMDEH03, Jamaica; 2Department of Biochemistry, Michael Okpara University of Agriculture, Umudike 440109, Nigeria; 3Department of Ophthalmology and Visual Sciences, The University of British Columbia, Vancouver, BC V5Z 0A6, Canada

**Keywords:** diabetic complications, phytochemicals, polyphenols, oxidative stress, diabetic nephropathy, diabetic neuropathy, diabetic retinopathy, antioxidants

## Abstract

The global prevalence of diabetes continues to increase partly due to rapid urbanization and an increase in the aging population. Consequently, this is associated with a parallel increase in the prevalence of diabetic vascular complications which significantly worsen the burden of diabetes. For these diabetic vascular complications, there is still an unmet need for safe and effective alternative/adjuvant therapeutic interventions. There is also an increasing urge for therapeutic options to come from natural products such as plants. Hyperglycemia-induced oxidative stress is central to the development of diabetes and diabetic complications. Furthermore, oxidative stress-induced inflammation and insulin resistance are central to endothelial damage and the progression of diabetic complications. Human and animal studies have shown that polyphenols could reduce oxidative stress, hyperglycemia, and prevent diabetic complications including diabetic retinopathy, diabetic nephropathy, and diabetic peripheral neuropathy. Part of the therapeutic effects of polyphenols is attributed to their modulatory effect on endogenous antioxidant systems. This review attempts to summarize the established effects of polyphenols on endogenous antioxidant systems from the literature. Moreover, potential therapeutic strategies for harnessing the potential benefits of polyphenols for diabetic vascular complications are also discussed.

## 1. Introduction

The change in nutrition habits due to rapid urbanization and an increase in the ageing population have contributed to an increase in the global prevalence of diabetes [1,2]. Consequently, diabetes is now a growing health concern, as the population of adults (aged 20 to 79) living with diabetes was estimated to increase from 463 to 536.6 million people between 2019 and 2021. An increment of a further 46%, to an average of 783.2 million diabetic adults, has been estimated to occur up to 2045 [3]. The latter increased level is projected to be significantly higher in middle-income countries compared to underdeveloped and developed nations [3].

Diabetes is the root cause of several macrovascular and microvascular complications. Macrovascular complications associated with diabetes include atherosclerosis and stroke; and microvascular complications include diabetic retinopathy, nephropathy, and neuropathy. Generally, diabetes vascular complications effectuate the majority of the burden of diabetes. The characteristics of diabetic complications may vary from context to context—in high-income countries, recent studies suggest a decline in hospitalization and death toll among older people with severe diabetic complications in the last 30 years [2,4]. However, in low- and middle-income countries (LMIC) such as Brazil, Ghana, Columbia, and China, and also among young adults, an increase in the prevalence of severe diabetic complications was observed [2]. This trend in LMIC was associated with healthcare deficiencies [2].

In this respect, especially of LMIC, a parallel buildup of diabetes vascular complications is set to have huge economic implications for these countries and the patients living with diabetes in these regions [1,2]. For instance, in North America and Caribbean (NAC) region, the cost of care for diabetes-related health conditions was estimated to be USD 414.5 billion. In contrast, South East Asian countries such as Cambodia, Myanmar which are LMIC, have double the population of diabetic patients compared to NAC regions and were estimated to spend USD 10.1 billion in 2021 [3], and in 2016 the cost of diabetes care in Africa was evaluated to be over USD 4 billion per annum [1]. Moreover, health expenditure increases parallelly by diabetes complications [1]. Furthermore, as a result of the financial burden and physical disabilities associated with diabetic complications, they significantly reduce the quality of life of those living with them [5]. In light of the growth of diabetes and the associated socio-economic impact of diabetes complications, interventions targeting improved therapy and lifestyle of diabetic patients are urgently needed.

Phytochemicals are secondary chemical metabolites that occur naturally in fruits, vegetables, and herbs that are known to possess different structures and can promote health benefits in humans [6]. Phytochemicals are critical in plant growth, development, and protection, including protection from harmful insects and microorganisms as well as extreme temperature and ultraviolet light exposure [7].

Phytochemicals are categorized principally based on their structures as alkaloids, flavonoids, carotenoids, phenolics, terpenoids, steroids, and essential oils [8,9,10]. Several phytochemicals are recognized as the bioactive components of commonly available medicines, examples including salicylates sourced from willow bark which is present in aspirin which is used as pain and fever relief [11,12], quinine found in cinchona bark and used to treat malaria [13]. The largest group of phytochemicals in plants are the polyphenols (e.g., flavanones, flavonoids) known for their free radical scavenging activity. Free radical scavenging activity of flavonoids, for example, is dependent on the number and position of hydroxyl groups on aromatic rings and a C ring with a double bond conjugated with a 4-oxo function, which confers on them their reducing capacity [14,15].

While phytochemicals provide the surest and quickest pathway to the discovery and development of new drugs, they also have some drawbacks which continue to restrict their overall acceptability. Challenges include the lack of a standardized dosage regimen, the preparation, drug–herb interactions, and low bioavailability, for example, clearance of oral contraceptives is increased by co-ingestion with St John’s wort [16]. Therefore, research into ways to maximize phytochemicals for therapy is required.

The purpose of the current review is to discuss the body of literature on diabetic micro-and macrovascular complications with a focus on: (1) the roles of oxidative stress (2) the activation of endogenous antioxidant defenses by exogenous phytochemicals, and (3) the therapeutic potentials of phytochemicals as alternative/adjuvant options.

## 2. Diabetic Complications

Due to the progressive nature of diabetes, it eventually leads to complications which are mainly microvascular and macrovascular. These complications have intense consequences on the anatomy, structure, and function of several cells, tissues, and organs, and consequently on the patient’s well-being status [17].

### 2.1. Diabetic Retinopathy (DR)

DR is the most common microvascular complication of diabetes that can affect the retinal metabolism, physiology, and microvasculature of the eyes, it is also determined recently as the prominent cause of blindness in the working population [18]. The impact of DR on normal vision varies broadly depending on the stage of DR and the earliest stage of DR may be asymptomatic [19,20,21,22]. General symptoms of DR which usually develop gradually could include temporarily blurred vision, floaters, and flashes of light or blind spots in the field of vision, and gradual or sudden loss of vision [23,24].

Generally, DR is classically categorized into two major classes: non-proliferative DR (NPDR) and proliferative DR (PDR). NPDR symptoms are defects of the retinal vasculature including hemorrhages such as dot hemorrhages and blot hemorrhages [25,26], hard exudation [26,27], cotton wool spots [25,26,27,28], microaneurysms [25,26,27], and vascular leakage [27,28]. The major PDR hallmarks are related to pathological retinal angiogenesis which involves the growth of new abnormal blood vessels from the pre-existing vascular network, and pathological retinal neovascularization which encompasses the development of new blood vessels by de novo formation (vasculogenesis) and angiogenesis [26,29,30,31,32,33]. The progression from NPDR to PDR is characterized by ischemia, hypoxia, and the increased expression of proangiogenic growth factors including vascular endothelial growth factor (VEGF), fibroblast growth factor-2 (FGF-2), platelet-derived growth factor (PDGF), and angiopoietin-2 (Ang-2) which activate the growth of aberrant retina blood vessels which can protrude into the preretinal space [30,31,34].

Apart from the clinical features already discussed, other hallmark pathological changes in DR include pericyte loss, microglial activation, modifications to macroglial functions (Müller cells and astrocytes), thickening of the basement membrane, retinal leukocyte adhesion, blood–retinal barrier (BRB) breakdown, leukostasis, capillary nonperfusion, and capillary endothelial cell injury and death [25,26,27]; and degeneration of retinal neurons [35,36].

Prevailing recommended treatments for DR are laser pan-retinal photocoagulation (PRP) [37,38], vitreoretinal surgery [27], and anti-VEGF intravitreal injections [39,40]. Although the aforesaid treatments may slow down the progression of DR towards blindness, they are not effective in tackling every instance of the disease and have significant side effects [38,41,42,43,44,45]; therefore, it becomes imperative to find exigent alternatives or adjuvant therapeutic options to prevent DR or slow down its progression towards blindness.

### 2.2. Diabetic Peripheral Neuropathy (DPN)

DPN is a microvascular diabetic complication that is characterized by peripheral nerve dysfunction. DPN is a significant cause of morbidity in diabetic patients and about half of all diabetic individuals suffer from this complication [46]. In the clinic, the manifestations of DPN include painful neuropathic sensations and insensitivity, the latter significantly increases the risk of unintentional injuries, burns, and foot ulcers, which potentially leads to non-traumatic amputation and a significant reduction in the health status of patients [47,48]. Once DPN initiates, it would be irreversible, but progression can be slowed down [49].

The Toronto Diabetic Neuropathy Expert group categorized DPN into three major classes: (I) possible DPN, wherein any of the following signs or symptoms may be present: symptoms–decreased sensation, positive neuropathic sensory symptoms (e.g., “asleep numbness”, prickling or stabbing, burning or aching pain) that predominantly occur in the toes, feet, or legs, signs—a symmetric decrease in distal sensation or unequivocally decreased or absent ankle reflexes; (II) probable DPN, wherein two or more of the following signs and symptoms of neuropathy are present: neuropathic symptoms, decreased distal sensation, or unequivocally decreased or absent ankle reflexes; and (III) confirmed DPN, wherein nerve conduction is anomalous and a sign or symptom of neuropathy is present [47].

DPN is currently managed by tight glycemic control focusing on reducing HbA1c, however, for many patients, it is challenging to accomplish glycemic control [48]. The pain symptoms of DPN are managed with pharmacological agents including anticonvulsants (e.g., pregabalin and gabapentin) as the first-line pain therapy, serotonin–norepinephrine reuptake inhibitors (e.g., duloxetine and venlafaxine), secondary amine tricyclic antidepressants (e.g., nortriptyline and desipramine) and opioid agonists (e.g., oxycodone and methadone) [48,49].

### 2.3. Diabetic Nephropathy (DN)

DN is another microvascular diabetic complication that is the global leading cause of end-stage renal disease (ESRD) and dialysis, constituting about 40% of total patients who need renal replacement therapy [50,51]. The mortality rate is about 30 times higher in DN patients compared with other diabetic patients without DN [52]. Glomerular basement membrane (GBM) thickening, mesangial matrix expansion, proteinuria (classically manifested as albuminuria), the development of characteristic Kimmelstiel–Wilson nodules, and progressive regression in glomerular filtration rate (GFR) are characteristic features of DN [53,54].

DN is classically categorized into five stages based on urinary albumin excretion (UAE), GFR, and blood pressure (BP) [52,54]: (1) glomerular hyperfiltration wherein, GFR is normal “>90 mL/min/1.73 m^2^” or increased, UAE value is <30 mg/day and BP is normotensive; (2) silent stage, wherein, GFR is normal. UAE rate is <30 mg/day and BP is ±hypertensive along with thickened basement membrane; (3) incipient nephropathy with GFR of <60 mL/min/1.73 m^2^, microalbuminuria (UAE of 30–300 mg/day; the earliest clinically detectable sign), and BP of ±hypertensive; (4) overt nephropathy, wherein, GFR is <30 mL/min/1.73 m^2^, macroalbuminuria (UAE of >300 mg/day) and hypertensive BP are present; and ultimately, (5) ESRD, wherein, GFR value is <15 mL/min/1.73 m^2^, with conditions of hypertensive BP and macroalbuminuria in which usually dialysis or transplantation is required [54].

Currently, tight glycemic and BP control, and the inhibition of the renin–angiotensin-aldosterone system (RAAS) via angiotensin-converting enzyme inhibitors or angiotensin II receptor blockers are the central approaches of DN therapy [52,54]. However, these management approaches have limitations in preventing DN progression towards ESRD, hence, effective therapies are urgently needed [54].

### 2.4. Diabetes-Induced Cardiovascular and Cerebrovascular Diseases

Diabetics are at a greater risk of both cardiovascular and cerebrovascular diseases including atherosclerosis, cardiomyopathy, and stroke, which significantly increase morbidity and mortality in these patients [55,56,57,58].

Atherosclerosis is a macrovascular disease that is the primary cause of various heart diseases and stroke; diabetes-induced atherosclerosis is a significant cause of morbidity amongst diabetics [56,59]. It manifests as intimal thickening, inflammation, and narrowing of arteries by the build-up of plaques [59]. The Insulin Resistance Atherosclerosis (IRAS) study [56] has shown that compared with non-diabetics, diabetics have an increased rate of progression of carotid atherosclerosis, with diabetics having twice the mean progression of intimal-medial thickness in both the common carotid artery and the internal carotid artery.

Diabetic cardiomyopathy is the diabetes-induced functional and structural change of the myocardium in the absence of other risk factors including hypertension, coronary artery disease, and significant valvular disease [55]. The lack of specific standardized guidelines governing both the diagnosis and treatment of diabetic cardiomyopathy makes a definitive diagnosis, and treatment problematic [60]. Clinical pathological changes include cardiac remodeling, diastolic dysfunction, myocardial fibrosis, cardiac stiffness, impaired calcium handling, increased atrial filling, enlargement, and increased left ventricular end-diastolic pressure [55,61].

Stroke (or cerebrovascular accident) emerges due to the combination of extracranial carotid artery disease and intracranial vessel diseases [62]. The risk of stroke is increased by 36% in diabetics compared with non-diabetics [63] and one-third of all stroke patients are diabetics [64]. There are two main forms of stroke that may feature in diabetes, ischemic and hemorrhagic stroke, with both having the potential of significantly increasing morbidity and mortality amongst diabetics [65]; and in the clinic, stroke is manifested as asymptomatic carotid artery occlusion or cerebral small vessel disease to transitory ischemic attack [62].

Strict glycemic and lipid control as well as management of hypertension are essential for managing both diabetes-induced cardiovascular and cerebrovascular diseases; furthermore, pharmacological interventions for diabetes-induced cardiovascular diseases including glucagon-like peptide 1 (GLP-1) receptor agonists, agonists of peroxisome proliferator-activated receptor gamma-γ (PPAR-γ) (e.g., pioglitazone), and the pharmacological inhibition of sodium-glucose cotransporter type 2 (SGLT2) and RAAS (e.g., by an angiotensin-converting enzyme (ACE) inhibitor or an angiotensin-receptor blocker) have shown clinical benefits [60,66,67,68]. Pharmacological interventions that can prevent stroke in diabetics include pioglitazone and anti-platelet drugs, such as, clopidogrel and aspirin; however, to reduce the combined risk of ischemic stroke, myocardial infarction or vascular death, clopidogrel may be more effective than aspirin [68,69,70]. Although these pharmacological interventions are available, they are inadequate for most diabetics and novel alternative adjuvant treatment options are needed.

## 3. Implications of Oxidative Stress in Diabetic Complications

Insulin deficiency due to the destruction/deficiency of β-cells is the main trigger of hyperglycemia in type 1 diabetes. A combination of factors such as hormonal and autoimmune dysfunction and environmental factors are associated with the destruction of β-cells [71]. In type 2 diabetes, hyperglycemia occurs due to impaired glucose uptake in skeletal muscles, deranged hormonal regulation of glucose metabolism, and impaired β-cell secretory function [72]. In essence, hyperglycemia due to β-cell dysfunction is common in diabetes. The progression of diabetes and the development of vascular complications stem from hyperglycemia-induced β-cell and vascular damage. This damage is known to be associated with reactive oxygen and nitrogen species (e.g., superoxide ion and nitric oxide).

Although various processes are implicated in the pathophysiology of diabetic retinopathy, five major metabolic pathways are prevalently accepted to significantly contribute to the pathophysiology of diabetic complications: formation and accumulation of AGEs (advanced glycation end-products), increased expression of the receptor for AGEs (RAGE) and its activating ligands, activation of protein kinase C isoforms (PKCs), increased flux via the hexosamine pathway, and increased polyol pathway flux [73,74,75,76,77].

The accumulation of AGEs and their interaction with RAGE can influence the permeability and self-regulation of vascular endothelial cells, inducing inflammation and releasing various proangiogenic cytokines and growth factors [76,78]. In DR, AGE accumulation and RAGE activation mediate the death of retinal pericytes, endothelial dysfunction, BRB breakdown, activation of NF-κB, and the induction of VEGF [78,79]. In diabetic complications, hyperglycemia promotes the production of diacylglycerol (DAG), an important co-factor of PKC-α, β, and δ isoforms [75,76,79]. PKC–DAG interactions are observed in the retina, glomerulus, heart, and aorta where they may promote the remodeling of the basement membrane and consequently promote vascular dysfunction in the aforementioned tissues [76,80,81,82].

The hexosamine pathway functions as an alternative pathway to the normal glycolysis pathway in the hyperglycemic milieu [77]. As a result, excessive uridine diphosphate N-acetylglucosamine (UDP-GlcNAc), the end-product of this pathway, is produced which can form glycolipids, glycoproteins, and proteoglycans which increases the burden of oxidative stress and worsen diabetic complications in various organs including the eyes and the heart [78,83,84,85]. In hyperglycemic conditions, one way that the activation of the polyol pathway contributes to the pathophysiology of diabetic complications is via increased aldose reductase (AR) activity which metabolizes excess intracellular glucose into sorbitol using NADPH as a cofactor and producing NADH [77,78,86]. As sorbitol accumulates within cells, osmotic damage and oxidative stress increase, and eventually, sorbitol is oxidized to fructose by sorbitol dehydrogenase (SDH) [78,86]. In DR, the accumulation of sorbitol and fructose in the retina correlates with increased oxidative stress, neuronal apoptosis, and astrocytes and Müller cell abnormalities [86,87]. The suppression of the polyol pathway has also been shown to impede the progression of experimental DPN, and clinical DPN, DN, and DR [88,89].

These pathways are major reactive oxygen species (ROS) pathways thereby key contributors to increased oxidative stress, a crucial phenomenon in the pathogenesis of diabetic complications [74]. Furthermore, these five pathways lead to elevated inflammation, inflammatory mediators, angiogenesis, and vascular anomalies [73,74,75]. However, of importance is the possibility that these pathways are not independent of themselves but likely stimulate pathological changes in diabetic complications via a complex interplay [75]. In fact, Brownlee [75] has provided multiple lines of evidence that these five pathways are controlled by a unified upstream event—the overproduction of mitochondrial superoxide by the mitochondrial electron transport chain.

### Hyperglycemia and Oxidative Stress

Glucose-dependent insulin secretion relies on calcium ion influx into β-cells. This voltage-dependent calcium ion influx is triggered by an increased ATP:ADP ratio due to rising glucose intracellular concentrations. In physiological conditions, the intracellular ATP:ADP ratio increases transiently and returns back. However, hyperglycemic conditions cause an excessive reduction in free ADP concentrations and an increase in the production of reducing equivalents (NADH and FADH_2_). This results in an increase in the availability of reducing equivalents and accelerated oxidative phosphorylation. Consequently, there is an excess of mitochondrial reductants such as coenzyme Q due to the lack of sufficient ADP for ATP formation. In this condition, the reduction in available oxygen occurs at a faster rate than ATP formation and ROS production ensues [90].

Endogenous antioxidant systems such as thioredoxin, glutathione, peroxidases, and the like (to be discussed in later sections) maintain ROS within optimal concentrations. However, in hyperglycemic conditions, antioxidant systems become dysfunctional and ROS-dependent signaling pathways are activated [91]. Furthermore, ROS generation from mitochondria and nicotinamide adenine dinucleotide phosphate (NADPH) oxidases (NOX) is implicated in insulin signaling. NOX1 is proposed to be upregulated in hyperglycemic conditions possibly due to increased diacyl glyceride-induced protein kinase C (PKC) levels. Consequently, excessive NOX activity causes uncoupling of endothelial nitric oxide synthase (eNOS) and increased production of peroxynitrite and superoxide [92,93].

ROS overproduction is associated with the activation of various pro-inflammatory and growth pathways. For example, ROS mitochondrial overproduction has been associated with reduced mammalian target of rapamycin (mTOR) activity. In chronic hyperglycemic conditions, reduced mTOR activity has been associated with 5′ adenosine monophosphate-activated protein kinase (AMPK) -induced increase in ROS production [94] and increased expression of thioredoxin-interacting protein [95]; all of which result in reduced β-cell differentiation and function [96]. Furthermore, increased ROS production has been associated with a reduction in protein kinase B (AKT)/mTOR activation leading to β-cell-induced apoptosis [97].

In the same vein, the AKT/mTOR pathway is associated with insulin signaling and activation of this pathway has been associated with improved survival of cardiomyocytes in diabetes-related myocardial infarction [98]. However, insulin resistance is proposed to be induced by mitochondrial ROS generation [99] and parallel inhibition of the phosphatidylinositol 3-kinase (PI3K)/AKT insulin signaling pathway [100]. Therefore, mitochondrial ROS generation is implicated in insulin resistance [99] and cardiac hypertrophy via mTOR-dependent mechanisms in diabetic conditions [98,101]. Furthermore, inhibition of insulin receptor substrate (IRS) via ROS inactivation of the PI3K/AKT pathway leads to inhibition of both glucose transporter 4 (GLUT4) translocation and glucose reabsorption in the skeletal muscles [99].

In essence, during over-nutrition states, a shift in redox status is implicated in insulin resistance and this results in impaired response to insulin in the skeletal/cardiac muscle and hepatic tissues with the characteristic dysfunctional control of glucose uptake and uncontrolled gluconeogenesis. In line with this, it is proposed that ROS-induced insulin resistance plays a major role in endothelial damage [73,102] and cardiovascular diabetic complications [73,93]. Overall, ROS modulation of insulin signaling and growth pathways such as the AKT/PI3K and mTOR pathways contribute to the progression of diabetes and the development of diabetic vascular complications.

In addition, ROS contributes to the progression of diabetes and the development of diabetic vascular complications by eliciting inflammation. For instance, downregulation of the mTOR signaling pathway via AMPK-dependent mechanisms has been shown to protect cardiomyocytes and skeletal muscles from inflammation [103,104]. Moreover, ROS is implicated in activation of redox-dependent NF-kB activation via PKC-dependent mechanisms [92,105]; redox-dependent serine/threonine phosphorylation and activation of JUN amino-terminal kinase (JNK)/extracellular signal-regulated kinase (ERK) pathways [105]. ROS activation of pro-inflammatory pathways is implicated in β-cell dysfunction and apoptosis; and endothelial damage. This ultimately causes the progression and development of cardiovascular complications [106].

## 4. Enzymatic Endogenous Antioxidants and Diabetic Complications

The endogenous antioxidant system consists of non-enzymatic and enzymatic antioxidants. Enzymatic antioxidants include superoxide dismutase (SOD) which can include metalloproteins such as manganese, copper/zinc SOD (MnSOD, Cu/ZnSOD), catalase (CAT), heme-oxygenases (HO), paraoxonase-1 (PON1) and the thioredoxins and glutaredoxins system which include: thioredoxin (TRx) and glutaredoxin (GRx), TRx reductase (TrxR), glutathione reductase (GR), glutathione (GSH), and NAD(P)H. Other Trx/Grx-dependent systems include glutathione peroxidase (GPx) and thioredoxin peroxidase (TrxP) [107] (Figure 1). These are the common antioxidant defense systems in the vascular wall to counteract ROS generation [108]. Non-enzymatic endogenous antioxidants include albumin, ceruloplasmin, and metal-binding proteins [109]. The transcription and expression of endogenous antioxidants are regulated by the nuclear factor erythroid 2-related factor 2/antioxidant response element (Nrf-2/ARE) system [110]. Generally, endogenous antioxidants diffuse reactive oxygen species, via redox reactions. In essence, mitochondrial enzymes such as MnSOD convert superoxide ion to hydroperoxide (Figure 1). Hydrogen peroxide in the presence of intracellular iron (Fe^2+^) forms hydroxyl ions (Figure 1) which oxidize proteins and cell membranes which can lead to deranged cellular signaling and apoptosis. However, TRx/GRx systems as shown in Figure 1, reduce hydroxyl ions and hydrogen peroxide thereby maintaining cellular redox homeostasis and cell viability [111]. The next paragraphs of this review will focus mostly on enzymatic endogenous antioxidant systems and their implications in diabetic complications.

### 4.1. Enzymatic Endogenous Systems Implicated in Diabetic Retinopathy

As mentioned earlier, oxidative stress due to a compromised antioxidant system (Figure 1) is central to the development of diabetic complications. In the retina, the formation of glycosylated proteins leads to the generation of free radicals and causes oxidative tissue damage due to glutathione (GSH) depletion [112]. Moreover, reduced activity of Nrf-2 and a parallel reduction in MnSOD, GPx, CAT, and mitochondrial function are suggested to occur in the retina in diabetic conditions [113]. Animal studies have shown that increased SOD3 (an isoform of extracellular SOD) prevented retinal muller cell activation and pericyte dysfunction [114]. Furthermore, extracellular SOD, CAT, and GPx are reduced in patients with DR; consequently, they are suggested to be potential biomarkers for DR [115,116]. Moreover, thioredoxin-interacting protein (TXNIP), an endogenous thioredoxin (Trx) inhibitor, works by inhibiting Trx reducing capacity and promoting cellular oxidative stress [117]. TXNIP is upregulated in diabetes and plays an important role in the pathophysiology of DR [118]. Taken together, the regulation of endogenous antioxidant enzymes in the diabetic retina is important in the management of DR.

### 4.2. Endogenous Antioxidants in Diabetic Nephropathy (DN)

The development of DN is attributed mostly to oxidative stress (OS) [119]. Some intracellular components, including the endoplasmic reticulum, peroxisomes, and mitochondria, as well as particular enzyme clusters, such as NADPH oxidases (NOX), are involved in the formation of ROS in kidney cells [120]. Increased ROS levels are caused by either increased production, a failure of antioxidant mechanisms, or a combination of the two [121]. OS causes inflammation, kidney scarring, and other pathophysiological changes, resulting in further glomerular filtration rate (GFR) decline and DN progression [122]. Podocyte damage, proteinuria, and tubulointerstitial fibrosis are also linked to OS [123]. Trx, CAT, cytochrome c oxidase, SOD, and GPx perform antioxidant defense in the kidney [124].

SOD is the first line of enzymatic antioxidant defense in the kidney [125]. The findings of studies on SOD in diabetes-induced chronic kidney disease (CKD) patients are contradictory and challenging to interpret. Some researchers discovered no significant difference in SOD levels between CKD patients and controls [126]. Others discovered lower SOD activity in hemodialysis and peritoneal dialysis patients when compared to controls [127]. However, in animal studies, SOD1-transgenic db/db mice and STZ-treated SOD1-transgenic mice had lower albuminuria, transforming growth factor (TGF)-β1, and collagen IV expression, as well as mesangial matrix expansion and lower oxidative stress markers than control diabetic mice [128]. SOD2 deficiency has been linked to worsening renal dysfunction, tubulointerstitial fibrosis, inflammation, and apoptosis [129] while SOD3 expression was significantly increased in the glomerulus and tubular area of db/db mice after supplementation with recombinant human SOD3. Recombinant human SOD3 supplements improved diabetic nephropathy by preventing ROS and the phosphorylation of the ERK pathway [130].

Hong et al. [131] also reported on the benefit of targeting extracellular SOD as a treatment for type 2 DN through intrarenal AMPK-PGC-1α-Nrf2 and AMPK-FoxOs signaling. Extracellular SOD increased phosphorylation of AMPK, activation of peroxisome proliferative-activated receptor γ coactivator 1α (PGC-1α), and dephosphorylation of forkhead box O transcription factor (FoxO)1 and FoxO3a. The protective effects were attributed to enhanced nuclear translocation of Nrf2 and subsequently increased expression of NADPH dehydrogenase 1 and HO-1 [131].

Furthermore, in STZ-treated diabetic mice and db/db mice, proximal tubule-specific overexpression of CAT in DN reduced angiotensinogen, p53, and proapoptotic Bcl-2 associated X-protein (BAX) gene expression and decreased renal ROS generation and tubulointerstitial fibrosis [132,133]. This suggests a role of CAT in DN. In addition, glomerular GPx expression was lower in diabetic rats than in the normal control rats, and plasma and urine GPx levels were significantly lower in patients with diabetic glomerulosclerosis than in patients without glomerulosclerosis [134].

Wang et al. [135] reported on the role of kidney GPx4 expression level in DN progression. The GPx4 expression level was significantly lower in DN patients than in healthy controls and was associated with disease severity and progression [135]. Similarly, mice with DN showed significant changes in ferroptosis-associated markers such as increased expression levels of acyl-CoA synthetase long-chain family member 4 (ACSL4) and decreased expression levels of GPx4 [136]. Hence, the endogenous antioxidant systems CAT, SOD, and GPx are implicated in DN, and their upregulation may provide some benefit to DN patients.

### 4.3. Endogenous Antioxidants in Diabetic Cardiomyopathy

The endogenous antioxidants present in the heart protect against ROS toxicity. They include antioxidant enzymes such as SOD, CAT, GPx, GST, and glucose-6-phosphate dehydrogenase. ROS, such as superoxide anion, hydroxyl radicals, and hydrogen peroxide, play a role in mediating oxidative reactions in the development of diabetic cardiovascular complications [137,138]. Indeed, increased levels of malondialdehyde, thiobarbituric acid reactive substances (TBARS), hydroxynonenal (HNE), 8-hydroxyguanosine, 8-hydroxy-2′-deoxyguanosine, and 8-iso-prostaglandin F2α are biomarkers of chronic oxidative stress in diabetic cardiomyopathy [139].

Cardiomyocyte enlargement, myocardial interstitial and perivascular fibrosis, inflammation, apoptosis, and elevated markers of oxidative stress are the main pathological characteristics of diabetic cardiomyopathy [140]. Apoptosis has been found to be significantly increased in the myocardium of diabetic subjects [141]. Overexpression of the mitochondrial tetrameric mnSOD or catalase in the diabetic heart enhanced respiration, shielded cardiac mitochondria from oxidative damage, and restored bulk in diabetic mitochondria [142].

Overexpression of GPx resulted in an improvement in left ventricular diastolic function, and a reduction in cardiomyocyte hypertrophy, interstitial fibrosis, and myocardial apoptosis in mice with streptozotocin-induced diabetes [143]. Diabetic mice failed to develop cardiomyopathy when GPx was overexpressed under the control of a housekeeping gene promoter [143]. Many natural and synthetic Nrf2 activators have been shown in animal models to be effective in preventing diabetic cardiomyopathy [144]. In view of this, it can be inferred that an altered endogenous antioxidant system is implicated in the pathophysiology of diabetic cardiomyopathy and its upregulation might prove beneficial in managing the disease.

### 4.4. Endogenous Antioxidants in Cerebrovascular Diabetic Complications

The maintenance of the brain against oxidative stress involves both enzymatic and non-enzymatic antioxidants [145]. In diabetic cerebrovascular disorders including stroke and cognitive impairment, ROS increases the susceptibility to neuronal damage and functional decline via brain oxidation [146].

Endothelial dysfunction characterizes ischemic stroke with an increase in ROS and lower levels of endogenous antioxidants in humans [147,148]. Both stroke patients and animal models of stroke with middle cerebral artery occlusion (MCAO) have been shown to exhibit increased protein and lipid oxidation [149]. Moreover, a considerable rise in malondialdehyde (MDA) levels was observed in patients with acute ischemic stroke 24 h and seven days after stroke, together with a decline in the levels of antioxidant enzymes [150].

Resveratrol improved endothelial dysfunction by reducing the release of cytokines that promote inflammation, such as TNF-α and IL-1 [151], lowering oxidative stress in a sirtuin1-dependent way [152] and promoting endogenous antioxidant enzymes such as SOD, CAT, and GPx [153]. Green tea extract and epigallocatechin-3-gallate (EGCG) reduced lipid peroxidation, which was increased after MCAO [154]. Additional improvements in neurological scores, decreased apoptotic neuronal death, increased GSH levels, decreased infarct size, and enhanced Nrf2 expression were observed after EGCG treatment [155] similar to the findings of Alfieri et al. on the benefit of Nrf2 activation in stroke injury [156].

Diabetes-induced cognitive impairment is characterized by deficits in intelligence, attention, psychomotor efficiency, information processing speed, cognitive flexibility, and visual perception [157]. Resveratrol hindered cognitive impairment induced by type 2 diabetes mellitus (T2DM) [157]. Resveratrol treatment prevented learning and memory decline in T2DM mice when compared to control mice [136]. Resveratrol also prevented T2DM-induced hippocampal neuronal destruction and synaptic ultrastructural damage and increased the expression of Nrf2 and its downstream genes which were decreased in the T2DM group compared to the control [136]. These findings were similar to the report of Han et al. on the prevention of cognitive impairment through activation of the Sirt1/Nrf2 signaling pathway in diabetic rats through antioxidative mechanisms [158].

A weakened endogenous antioxidant system is seen in various diabetes-induced cerebral complications including ischemic stroke and cognitive impairment, while upregulating the endogenous antioxidant system provided some benefits in experimental models.

## 5. Effects of Antioxidant Phytochemicals on Endogenous Antioxidant Pathways

### 5.1. Effects of Antioxidant Phytochemicals on Mitochondrial Uncoupling Protein 2 (UCP2) and Mitochondrial Function in Diabetic Complications

Mitochondrial ROS generation is central to oxidative stress induced by hyperglycemia and contributes to the development and progression of diabetic complications. Excessive mitochondrial ROS generation is associated with an increase in mitochondrial membrane potential. In fact, a reduction in membrane potential by mitochondrial uncoupling protein 2 (UCP2) is known to reduce ROS production and prevent cell apoptosis [159].

Caffeic acid phenylethyl ester (CAPE), a polyphenol has been shown to protect retinal cells against tert-butyl hydroperoxide induced oxidative stress. CAPE is known to have a similar structure to the 4-hydroxy-2E-nonenal natural inducer of UCPs. However, in this study [160], CAPE was observed to be less toxic than 4-hydroxy-2E-nonenal. It was also shown to effectively reduce mitochondrial membrane at lesser concentrations than 4-hydroxy-2E-nonenal. This activity of CAPE was inhibited by UCP2 siRNA and inhibitors of UCP2 without affecting mRNA levels of UCP although a minimum but significant increase in ATP production was observed [160]. This implies that, CAPE is similar to the endogenous UCP inducer-4-hydroxy-2E-nonenal: as it also reduces ROS production by reducing mitochondrial membrane potential via induction of UCP (Figure 2). Albeit, it has minimum effect on cellular energy metabolism and cell viability compared to 4-hydroxy-2E-nonenal.

Resveratrol is proposed to prevent arrhythmia in diabetic conditions by alteration of UCP2 levels [161]. The altered expression of UCP2 has been linked to peroxisome proliferator-activated receptor gamma coactivator1-alpha (PCG-1α), a regulator of mitochondrial biogenesis [162] which is known to both upregulate and indirectly downregulate the expression of UCP2 as negative feedback response.

Mitochondrial membrane potential is also regulated by the alteration of mitochondrial permeability via the formation of mitochondrial permeability transition pore (mPTP). The voltage-dependent anion channels (VDAC) and adenine nucleotide translocator (ANT) form the major channels for the exchange of ATP, proteins, water, and ions across the mitochondrial membrane. Opening or Closing of these channels is regulated by binding of anti/pro-apoptotic proteins (i.e., Bcl-2 /Bax), cyclophilin D, outer membrane transporter protein (18 kDa) (TSPO), hexokinase, and creatinine kinase [163]. Polyphenols are proposed to modulate mPTP by altering the interaction of these “regulatory” proteins with VDAC and ANT; or by directly binding to ANT as extensively reviewed by Naoi et al. [163]. For example, resveratrol prevents mitochondria membrane potential collapse by preferentially increasing the interaction of antiapoptotic protein Bcl-2 with VDAC in the outer membrane of the mitochondria. This causes inhibition of mPTP opening, preventing the release of cytochrome c and inhibition of apoptosis [163]. This effect could be beneficial in conditions such as myocardial infarction, which is characterized by the accumulation of mitochondrial ROS and cell death due to the opening of mPTP, leading to the collapse of mitochondrial membrane potential, the disruption of ATP production and induction of necroptosis [164].

Furthermore, anthocyanins (cyanidin-3-glucoside, delphinidin 3-O-glucoside, pelargonidin 3-O-glucoside) are reported to prevent the uncoupling of mitochondria in ischemic cardiac tissue. Skemiene et al. [165] observed that anthocyanins act as a substrate for mitochondria complex 1, reduce cytochrome c, and thereby preserve mitochondria function. A reduction in cytochrome c has been associated with the inhibition of caspase-dependent apoptosis and restoration of cardiac function in diabetic rats [166]. Taken together, polyphenols exert a beneficial effect against ROS-induced vascular damage by maintaining mitochondrial membrane potential via UCP2 and mPTP regulation; improving mitochondrial biogenesis, improving electron transport chain function; thereby preventing necroptosis and cell death [167,168].

### 5.2. Effects of Antioxidant Phytochemicals on Glutathione Peroxidase 4 (GPx4) and Coenzyme Q (CoQ) in Diabetic Complications and Ferroptosis

Ferroptosis is a programmed cell death that occurs due to iron-dependent lipid peroxidation eliciting necro-inflammatory processes [169]. Iron overload is associated with the progression of diabetes [170] and diabetic complications [171], likely due to hyperglycemia-induced expression of iron transporters in the intestines [172]. Iron propagates the formation of lipid peroxide and eventual cell death via the Fenton reaction, enzymatic autoxidation of polyunsaturated fatty acids, and lipoxygenase catalytic formation of arachidonoyl-(AA-)OOH-phosphatidylethanolamine/arachidonic acid [173].

Furthermore, lipid peroxides break down into hydroxy fatty acids or reactive aldehydes MDA [174]; the latter is a known marker of oxidative stress in type 2 diabetic patients [175]. GPx4 and coenzyme Q (CoQ) are two major antioxidants associated with reducing lipid peroxidation of mitochondrial and cellular membranes leading to ferroptosis [169,176].

As mentioned earlier, anthocyanins and quercetin are known to display “CoQ_1_ like” activity by acting as a complex 1 substrate during oxidative phosphorylation [165]. They may also bind lipid peroxides, thereby stabilizing the mitochondrial membrane during ischemia [165]. In addition, kaempferol and apigenin have been shown to increase CoQ biosynthesis in kidney cells [177] and resveratrol and para-coumarate serve as precursors for CoQ synthesis in human cells [178]. Based on the study by Fernández-del-Río et al. [177], it appears that flavonoid structure with a single hydroxyl group located on the C4′ in the B ring together with a pie bond between the A and C ring is important for the significant synthesis of CoQ. This is a common feature for resveratrol, kaempferol, and apigenin. In addition, the presence of these features and a hydroxyl group situated at C3 of the C ring elicits a much higher synthesis of CoQ_10_ as observed for kaempferol. Beyond CoQ synthesis, resveratrol was observed to increase mRNA expression of genes for CoQ synthesis and increase CoQ_10_ levels in the liver of high-fat diet mice; however, the change in CoQ_10_ levels was not significant [179].

Furthermore, kaempferol is reported to reduce the risk of diabetic complications in diabetic rats by decreasing lipid peroxidation and increasing glutathione/glutathione peroxidases in liver, kidney, and heart tissues [180], but the effect on CoQ levels is not reported. Polyphenols may contribute to CoQ levels and prevent lipid peroxidation and ferroptosis. However, further studies are required.

The role of GPx4 in ferroptosis and the implication for diabetic complications such as retinal damage; kidney damage, cardiomyopathy, and cognitive dysfunction is exemplified with evidence reviewed by Yang and Yang [176]. In brief, GPx4 is dependent on GSH availability for the reduction in lipid peroxides. GSH in turn is influenced by cysteine synthesis and xc^−^ transporter system, and NAD(P)H. The latter is affected by NOX activity which is implicated in cardiac injury [181] and diabetic cardiovascular diseases [182]. Therefore, inhibiting NOX activity, upregulating GSH/GPx4 activity, and reducing iron overload could be beneficial for preventing hyperglycemia-induced ferroptosis and related organ damage [176].

In line with the above, glabridin reduced genetic and protein expression of transferrin receptor 1, while increasing genetic and protein expression of GSH/GPx4 and xc^−^ transporter system. The study suggests that the reno-protective effect of glabridin was in part related to the reduction in iron load in the kidney and prevention of lipid peroxidation via upregulation of the GSH/GPx4 system [183].

Similarly, calycosin, an isoflavone, reduced markers for cellular labile iron pool and increased GPx4 in kidney cells exposed to hyperglycemic conditions [184]. In the kidneys of diabetic mice model, calycosin was proposed to inhibit hyperglycemia-induced injury by modulating ferroptosis [184]. In relation to diabetic cardiac diseases, naringenin was observed to reduce lipid peroxidation, and NOX2 expression in the cardiac tissue of diabetic mice, indicating potential ferroptosis mechanisms [185]. Moreover, astragaloside-IV is reported to reduce high-glucose-induced oxidative stress in retinal pigment epithelial cells by increasing cysteine synthesis and GPx4 expression via Nrf2-dependent mechanisms [186].

Prevention of vascular complications could also be attributed to the preservation of B-cell function. Cryptochlorogenic acid, a major active component of mulberry leaf, and quercetin are known to prevent lipid peroxidation and consequent ferroptosis by upregulating GPx4 and reducing the accumulation of iron in pancreatic cells in animal diabetic models and in vitro studies [187,188]. Taken together, polyphenols prevent ferroptosis by modulating GSH synthesis, iron intracellular transport, and upregulating the GPx4 antioxidant system (Figure 2), which could be beneficial for preventing the progression of diabetes and vascular complications.

**Figure 2 antioxidants-12-00123-f002:**
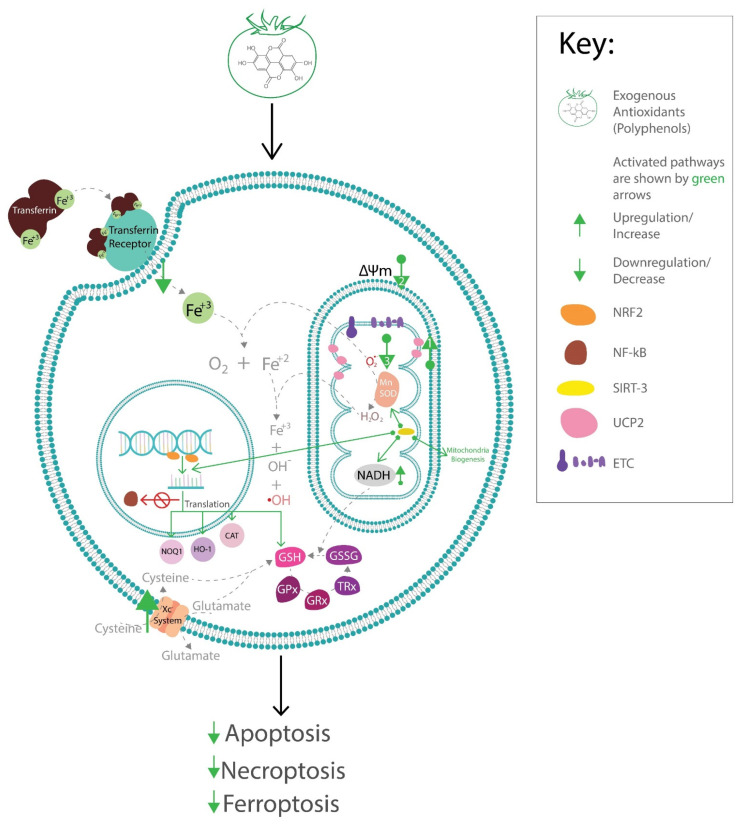
Summary of the effects of polyphenols on endogenous antioxidant systems. In hyperglycemia-induced oxidative stress, polyphenols increase UCP expression (1), reduces mitochondria membrane potential ΔΨm (2) which leads to reduction in O_2_^−^ (3); preventing mitochondrial damage and apoptosis/necroptosis. Polyphenols enhance Gpx/GSH/Trx function by increase in synthesis of GSH via increased activity of Xc^−^ transporter system, reduction in intracellular transport of Fe^2+^. Moreover, there is an increase in deacetylation activity of SIRT 3 which increases activity of isocitrate dehydrogenase activity (not shown) [189] and NADH leading to regeneration of GSH. Both the latter and former prevent ferroptosis. Polyphenols also enhance SOD activity via SIRT1/3-dependent mechanisms. Finally, increased SIRT3 activity also increases translation of several antioxidant enzymes including NOQ1, HO-1, and CAT via Nrf2-dependent mechanisms. Activation of Nrf2 inhibits NF-kB transcription activity, thereby preventing oxidative stress-induced inflammation which is associated with insulin resistance and endothelial damage.

### 5.3. SIRTs and Polyphenols

Silent information regulator proteins (SIRTs) are regulator proteins that couple cellular nutrient sensing to intracellular redox homeostasis via NAD^+^-dependent mechanisms. SIRTs belong to the class III histone deacetylases and exist as SIRT 1-7 isoforms. SIRTs are localized in the cytoplasm, mitochondria, and nucleus. They are involved with the regulation of redox signaling, transcription of Nrf2 with the expression of antioxidant enzymes, and mitochondrial biogenesis [189,190]. The functions of SIRTs are extensively reviewed in Singh et al. [189].

Resveratrol has been shown to activate SIRT1 activity thereby reducing oxidative stress and the progression of diabetes-induced kidney dysfunction in rats. SIRT1 activity reduced the acetylation of FOXO3a, a key regulator of MnSOD expression [189,191]. Upregulation of SIRT1 antioxidant activity by resveratrol is also associated with PGC-1α activation and the reduction in apoptosis with the improvement of mitochondria biogenesis [192]. Similarly, resveratrol is also shown to improve cardiomyopathy via the SIRT1/PGC-1α-dependent pathway [162]. Resveratrol is also reported to prevent mitochondrial membrane collapse by preventing mPTP pore opening. The suggested mechanism was related to SIRT3-dependent inhibition of cyclophilin-D interaction with ANT [193].

Furthermore, luteolin and fisetin were reported to increase the expression of SIRT1, 3, and 6 in human monocytic cells subjected to hyperglycemic conditions [194]. An increase in SIRT 1, 3, and 6 was associated with increased FOXO3a activity suggesting an effect of antioxidant enzymes [194]. In this respect, it can be said that resveratrol increases the activity of antioxidant systems (GSH, CAT, SOD) via SIRT1-dependent mechanisms. This implies that it could be useful for the prevention of diabetic kidney damage [195]. Interestingly, resveratrol [196], luteolin, and fisetin [194] are also observed to reduce NOX activity, suggesting the potential effect of preserving NADH for GSH antioxidant activity and related systems in the previous sections.

In light of the above, it can be inferred that polyphenols activate SIRTs which in turn upregulate enzyme antioxidants (Figure 2). Polyphenol activation of SIRTs also preserves mitochondrial function and induces mitochondrial biogenesis. This effect may also contribute to the prevention of ferroptosis by preserving GSH/GPx antioxidant activity (Figure 2).

### 5.4. Effects of Polyphenols on the Nuclear Factor (erythroid-derived 2)-like 2 (Nrf2) and Endogenous Antioxidant System

#### 5.4.1. Nrf2 Regulation

There are seven Nrf2 ECH homology domains (Neh 1-7) that regulate both the activation and inhibition of Nrf2 activity [197]. Under normal conditions, Nrf2 is targeted by Kelch-like ECH-associated protein 1 (KEAP-1) for cytosol degradation. KEAP-1 binding to Nrf2 leads to ubiquitination by Cul3-Rbx1-E3 ligase. This negative regulator of Nrf2 (KEAP-1) attaches to the amino-terminal of the Neh-2 domain containing DLG and ETGE as its binding motif through a double glycine repeat domain. Ubiquitination of Nrf2 leads to its proteasomal degradation. In essence, KEAP-1 inhibits Nrf2 transcriptional activity [198].

Nrf2 stability is negatively controlled by the Neh-6 domain (Serine rich) which acts independently of KEAP-1. DSGIS and DSAPGS, its conserved peptide motifs are recognized by β-transducing repeat-containing protein (β-TrCP). Glycogen synthase kinase -3β (GSK-3β)-mediated phosphorylation of the DSGIS motif causes β-TrCP to bind more effectively to Neh-6 and promotes S-phase kinase 1 (SKP1)-cullin1 (Cul1)-F-box protein 1 (SCF) ubiquitin ligase complex leading to Nrf2 proteasomal degradation [199]. Moreover, the Neh-7 domain through binding with retinoid x receptor (RXRα) mediates repression of Nrf2 transcriptional activity [200].

Excess ROS causes redox disequilibrium and Nrf2 is translocated to the nucleus after Nrf2-KEAP-1 is disassembled. Once inside the nucleus, the Nrf2 Neh-5 domain functions as a transcription factor and has a redox-sensitive nuclear export signal. Furthermore, the CNC-bZIP region on Neh-1 plays a crucial role in the binding of Nrf2 to DNA and the small musculoaponeurotic fibrosarcoma (smaf) proteins [201] allowing Nrf2 to bind to antioxidant response elements (AREs: enhancer sequences in the regulatory region of Nrf2 target genes). This ultimately leads to activation and the expression of antioxidant genes and iron genes involved in the stress response [110]. In addition, ubiquitination and phosphorylation of KEAP-1 by protein kinase C (PKC) phosphorylation also regulate Nrf2 activity [202]. Other KEAP-1-independent regulation of Nrf2 includes the antagonism of GSK-3β action by PI3k-/AKT signaling [197].

Nrf2 activators including phytochemicals can upregulate Nrf2 activity to maximize its antioxidant activity either through structural inhibition of KEAP-1, activation of the Nfe2 (nuclear factor, erythroid 2) gene transcription, micro-RNA induced degradation of KEAP-1, inhibition of proteasomal Nrf2 protein degradation and modulation of upstream regulators of Nrf2. The following paragraphs will discuss examples of polyphenols that activate Nrf2 in select diabetic complications.

#### 5.4.2. Nrf2 in Diabetic Cardiomyopathy

Naringenin, a compound with known antioxidant and anti-inflammatory effects, has been investigated for its potential benefits in STZ-induced diabetic cardiomyopathy. Naringenin decreased blood sugar levels along with a reduction in myocardial fibrosis and cardiomyocyte death [185]. Naringenin also significantly increased the expression of antioxidant enzymes and Nrf2, while inhibiting pro-inflammatory cytokines, reactive oxygen species levels, and nuclear factor kappa-B (NF-kB) expression [185]. Naringenin may control Nrf2 and NF-kB pathways to prevent diabetes-related cardiac damage by lowering oxidative stress and reducing inflammation [185].

Furthermore, garlic oil (diallyl trisulfide) boosted Nrf2 protein stability and nuclear translocation, protecting against hyperglycemia-induced ROS-mediated apoptosis by upregulating PI3K/AKT/Nrf2 pathways [203], in cardiomyocytes exposed to high glucose. This suggests that garlic oil may improve insulin sensitivity in cardiac myocytes via Nrf2-dependent mechanisms.

#### 5.4.3. Nrf2 in Diabetic Nephropathy

Nrf2 is elevated in diabetic kidneys in both humans and animals in the early stages of the disease while its capacity to translocate into the nucleus is decreased, indicating functional impairment of Nrf2 [204]. Consequently, impaired NrF2 activity in DN leads to the accumulation of peroxide radicals which harm kidney cells [205].

EGCG reduced DN by activating the Nrf2/ARE signaling pathway by downregulating KEAP-1 and disrupting the Nrf2-KEAP-1 interaction, thereby increasing nuclear Nrf2 [204]. Sun et al., 2017 confirmed these findings by reporting that EGCG increased Nrf2 expression to prevent diabetic nephropathy [206]. A 24-week EGCG treatment activated Nrf2 expression and function in wild-type mice without affecting KEAP-1 expression. Diabetes-induced renal oxidative damage, inflammation, fibrosis, and albuminuria were all reversed by Nrf2 activation, while deletion of the Nrf2 gene negated the effects of EGCG [206]. Taken together, the data suggest that EGCG could activate Nrf2 via the KEAP-1-dependent and independent pathway. 

Altamimi et al. [207] investigated the role of ellagic acid in the prevention of DN. Ellagic acid reduced NF-κβ, upregulated GSH, γGCL, and SOD, and increased nuclear translocation of Nrf2 while preventing renal damage, oxidative stress, inflammation, and apoptosis. Ellagic acid also inhibited KEAP-1 cytoplasmic expression and interaction with Nrf2, as well as increased AKT and GS3KB phosphorylation [207].

Sinapic acid, a polyphenolic metabolite, was found to have antioxidant and anti-inflammatory properties in DN by upregulating the Nrf2/HO-1 signaling pathway [208]. A similar effect has also been observed for astaxanthin in diabetic rats. In addition to upregulating the Nrf2/HO-1 signaling pathway, astaxanthin also increased SOD-1 and reduced fibronectin and collagen IV accumulation in the kidneys of diabetic rats. Astaxanthin also reduced MDA levels suggesting a reduction in oxidative stress in diabetic rats [209].

Paeonol (PA), a phenolic compound derived from the cortex mountain root bark, was tested for antioxidant activity in glomerular mesangial cells and STZ diabetic mice by Zhang et al. [210]. PA provided renoprotection by activating the Nrf2/ARE pathways via SIRT1, suggesting that PA slows the progression of diabetic renal fibrosis [210].

Hesperitin is also reported to provide renoprotection against DN by upregulating the Nrf2/ARE/glyoxylase pathway and augmenting γGCL, a target gene of Nrf2/ARE signaling [211].

These evidences suggest that polyphenols activate the endogenous antioxidant systems in DN via upregulation of Nrf2 activity by inhibiting its proteasomal degradation and improving its translocation to the nucleus and could provide an additional option in the pharmacotherapy of diabetic complications including DN.

#### 5.4.4. Nrf2 in Diabetic Retinopathy

Shi et al. [212] investigated the effects of rhaponticin, a polyphenol stilbene, in DR of diabetic rats. Rhaponticin caused upregulation of Nrf2, NQO-1, and HO-1, while MDA levels were significantly reduced, along with increased activity of SOD, CAT, and GPx. Other effects of rhaponticin include the downregulation of TNF-α, matrix metalloproteinase-2, and the upregulation of IL-10 and tissue inhibitor of metalloproteinase-1 (TIMP-1) in the retina. These findings demonstrate the importance of Nrf2 in maintaining redox homeostasis in the retina, as well as the ability of rhaponticin to prevent diabetic retinal changes via antioxidant, anti-inflammatory, and hypoglycemic effects [212].

Curcumin, a natural polyphenol significantly reduced retinal edema and outperformed insulin in terms of anti-photoreceptor apoptosis [213]. Curcumin alleviated compensatory Nrf2 pathway activation by directly reducing free radicals when combined with insulin, and it better maintained Nrf2 pathway homeostasis compared to insulin alone in the early stages of diabetes [213]. When combined with hypoglycemic agents, curcumin was beneficial in the management of diabetic complications [213]. Bucolo et al. also reported a similar effect of curcumin in high glucose-induced DR [214]. Antioxidant activity was linked to the activation of the Nrf2/HO-1 pathway through the ERK pathway [214]. This suggests antioxidant and anti-inflammatory effects of curcumin in DR.

Blueberry anthocyanin extract (BAE) protected human retinal capillary endothelial cells from high glucose injury [215]. BAE increased the levels of Nrf2 and HO-1 mRNA while decreasing vascular endothelial growth factor (VEGF) and increasing antioxidant enzymes such as GSH and GPx. The antioxidant effect was achieved by activating the Nrf2/HO-1 pathways [215]. A comparison of BAE and the anthocyanin constituents (malvidin, and malvidin glucosides) showed that malvidin-3-glucoside was the most effective for inhibiting high glucose-induced ROS production in human retinal capillary endothelial cells (HRCECs). Moreover, malvidin-3-glucoside did not inhibit NOX4 expression but increased CAT and SOD expression. Furthermore, BAE, malvidin, and glucosides inhibited VEGF, ICAM, and NF-κB expression [216]. Malvidin-3-glucoside and other anthocyanins in grape pomace were suggested to modify KEAP1/Nrf2 interaction and increase the expression of HO-1 and NQO1 antioxidants [217]. This suggests that malvidin-3-glucoside and the likes inhibit inflammation in endothelial cells by improving antioxidant enzymatic expression via Nrf2 signaling-dependent mechanisms.

Pterostilbene, a natural stilbene with documented antioxidant activity has shown protective activities against DR [218]. Pterostilbene conferred protection by reducing protein and lipid oxidative damage and restoring antioxidant enzyme activities, which it accomplished by activating the PI3K/Akt/GSK3/Nrf2 pathways [218]. Matos et al. [219] extensively reviewed the benefits of flavonoids in DR [219].

In summary, polyphenols can confer protection against oxidative stress via upregulating endogenous antioxidants through the Nrf2/KEAP-1 pathway in DR and could be pharmacologically targeted in the future management of DR.

#### 5.4.5. Nrf2 in Diabetic Peripheral Neuropathy

DPN pathogenesis is influenced by oxidative stress and neuroinflammation [220]. Phytochemicals have also been studied as Nrf2 activators in the management of DPN. Fisetin, a phytoflavonoid, has been shown in experimental DPN to protect from oxidative stress and neuroinflammation via NF-κB inhibition and Nrf2 upregulation [221]. Nrf2 increased the levels of antioxidant enzymes such as HO-1, NQO-1, and GCL [221].

Morin, a bioflavonoid with documented antioxidant capacity has been investigated for its benefits in experimental DPN [222]. Morin reduced high glucose-induced mitochondrial superoxide production, membrane depolarization, and ROS generation while increasing Nrf2 activity and inhibiting Nf-KB-mediated neuro-inflammation [222]. Rutin was also beneficial in the STZ model of DPN. When compared to DPN control rats, rutin combined with the cyclooxygenase 2 (COX-2) inhibitor nimesulide reversed oxidative damage as evidenced by increased levels of SOD, CAT, and GSH, as well as Nrf2/HO-1 activation in the sciatic nerve [223].

Overall, Nrf2 regulates the expression of several antioxidant enzymes and can be said to be central to increased endogenous antioxidant activity. Moreso, polyphenol antioxidant activity via modulation of Nrf2 signaling inhibits hyperglycemia-induced inflammation and apoptosis (Figure 2). This emphasizes the benefits of polyphenols for targeting the Nrf2 signaling pathway in the management of diabetic complications. Table 1 below highlights examples of polyphenols and their effect on antioxidants for the benefit of various diabetic complications.

#### 5.4.6. Polyphenol and Endogenous Antioxidants Effect

From the previous sections, the polyphenol effect on SIRT1/3 activity can be considered a signaling connector between ROS production, GPx function, and Nrf2 signaling. This is because they activate MnSOD, increase NADH which is crucial for GPx function and increase Nrf2 signaling. The latter then leads to the upregulation of other antioxidants and anti-inflammatory effects (Figure 2). Moreover, ROS overproduction is associated with insulin resistance [99,103], and polyphenol upregulation of endogenous antioxidants via Nrf2-dependent mechanisms can be said to be beneficial for insulin resistance by the upregulation of the PI3K/AKT pathway [218]. Polyphenol modulation of mTOR via the PI3K/AKT pathway could be beneficial for the prevention of cardiac hypertrophy and preserving β-cell function [97,98,101]. Therefore, targeting the Nrf2 pathway and the SIRTs could be beneficial for the prevention of insulin resistance and associated diabetic cardiovascular complications. Regulation of the Nrf2 pathway by microRNAs has been studied. microRNAs could also be modulated by polyphenols and can indirectly modulate antioxidant activity [224].

Other antioxidants such as paraoxonase are modulated by polyphenols. Glabridin, at levels lower than antioxidant concentrations, was shown to increase expression and transcription of PON2, in addition to MnSOD and CAT in hyperglycemic mice. The mechanism of action was suggested to be via protein binding with PON2 [225]. A similar study with curcumin-enriched yoghurts and metformin in diabetic rats showed that curcumin sustained the function of PON1 in diabetic conditions probably via preventing glycation of PON1 [226]. Taken together, glabridin and curcumin could prevent atherosclerosis by upregulating PON activity and reducing lipid peroxidation [225,226].

Glabridin was also shown to inhibit lipopolysaccharide-induced inflammation in THP-1 cells by downregulating the expression of inducible nitric oxide synthase (iNOS). Downregulation of iNOS and nitrotyrosine production was also observed in the liver of high-fat-fed mice. This suggests that glabridin could prevent inflammation associated with nutritional-induced oxidative stress (i.e., hyperglycemia) [227].

In relation to inflammation, polyphenols such as punicalagin, prevented nucleotide-binding oligomerization domain, leucine-rich repeat, and pyrin domain-containing protein 3 (NLRP3) induced pyroptosis (inflammation-induced cell death) by inhibiting NOX4 activity. This in turn reduces Trx-TXNIP dissociation and subsequent activation of NLRP3. The data suggest the role of polyphenol modulation of the Trx antioxidant system for pyroptosis in the development of DN [228].

Polyphenols could also indirectly exert antioxidant activity through highly lipophilic gut microbial-derived metabolites [229]. Therefore, polyphenol gut microbiota interactions contribute to systemic polyphenol antioxidant activity. Iatcu et al. [230] extensively reviewed the role of gut microbiota in various type-2 diabetes complications. In general, the proportion of Bifidobacterium spp. and Lactobacillus spp. to Enterococcus spp. and Clostridium spp. is reduced in patients with type 2 diabetes and complications.

Gallic acid is known to modulate gut microbiota in different conditions including stress and inflammatory bowel disease [231]. An evaluation of healthy dog fecal content revealed that long-term supplementation of gallic acid increased anti-inflammatory short-chain fatty acid concentrations. This increase was positively correlated with an increase in Erysipelotrichaceae_unclassified spp. and Faecalibacterium spp., and a decrease in the proliferation of pro-inflammatory Parasutterella spp.; suggesting a beneficial prebiotic and anti-inflammatory effect of gallic acid. In this study, gallic acid supplementation was also associated with increased levels of CAT and SOD [231].

In a similar study, olive oil enriched with polyphenols and thyme elicited an increase in Bifidobacterium spp. in humans. Furthermore, a reduction in Clostridium spp. was associated with a decrease in isolithocholic acid; the latter is associated with a reduced risk of cardiometabolic disease in chronic insomnia [232]. Furthermore, the prebiotic effect of olive oil enriched with polyphenols was associated with reduced oxidized low-density lipoproteins and increased microbial metabolites of procatechuric and hydroxytyrosol, suggesting a relationship of gut-derived metabolites and antioxidant activity [233].

In fact, specific gut microbiota-derived polyphenolic metabolites also retain the ability to modulate antioxidant systems. For example, in vitro studies on urolithins A and B (gut microbial metabolites derived from polyphenols) are known to preserve podocyte function by upregulation of Bcl-2 expression and nephrin expression. Urolithin A, specifically, upregulated Bcl-2 in high glucose conditions [229]. In vivo studies with urolithin A have been shown to prevent renal kidney injury in vivo via the Nrf2 signaling pathway [234]. These results suggest that polyphenols–gut microbiota interactions contribute to the modulatory effect of polyphenols on endogenous antioxidants.

Overall, polyphenols influence endogenous antioxidant systems via multiple mechanisms. A summary of the effects of polyphenols on some endogenous antioxidant systems is represented in Figure 2 below. 

## 6. Potential Therapeutic Strategies to Alleviate Diabetic Complications via the Activation of Endogenous Antioxidants by Exogenous Antioxidant Phytochemicals

One therapeutic strategy for harnessing the beneficial effects of polyphenols could be drug synthesis. For example, redesigning polyphenols to include imidazole derivatives [235,236] might prove a potential way to harness polyphenols for insulin resistance via Nrf2-dependent mechanisms. Structure-related activity of studies could assist with the identification of phenolic components with high affinity for inducing antioxidant systems. For instance, cyanidin is a potential drug candidate for antioxidant therapy as it was identified as a potent activator of SIRT6 [237]. Further drug discovery studies are required to harness this potential of cyanidin. For example, the chemical synthesis of cyanidin analogs coupled with in silico studies could facilitate the discovery of cyanidin analogs with optimized pharmacokinetic properties and less toxicity [238]; making it useful for therapy of diabetic complications. Taken together, further studies of chemically modified polyphenols and their effect on antioxidant systems are required.

Beyond synthetic drug designs, harnessing the health benefits of polyphenols in their natural form depends on their bioavailability. However, bioavailability is influenced by several factors such as food source, food processing, gut microbiome/hepatic biotransformation, and individual variability. Rodriguez-Mateos et al. [239] reported the bioavailability of polyphenols after the consumption of berry drinks compared to berry bun in healthy volunteers. Bioavailability and physiological effects on the endothelium were associated with heat processing and gut microbiome biotransformation—only the latter had an effect six hours post-consumption and was associated with unprocessed berry drink [239]. With respect to studies by Rodriguez-Mateos et al. [239], cooking may also improve the availability of polyphenols as reported in the study by Abukhabt et al. [240]. Heat processing causes a release of polyphenols from the food matrix and can increase absorption and colonic concentrations both of which are crucial for polyphenol bioactivity.

As discussed earlier, it is accepted knowledge that the gut microbiome is implicated in the therapeutic benefits of polyphenols. Gut microbiome dysbiosis is implicated in the disruption of intestinal tight junctions, a common pathological feature of aging and diabetes. Hidalgo-Liberona et al. [241] suggest that decreased intestinal integrity may reduce the bioavailability of gut microbiome-derived polyphenolic metabolites but increase the bioavailability of phase II metabolites. The implication of this could mean that when patients with compromised intestinal integrity (e.g., diabetic patients) consume flavonoid-rich metabolites, for instance, there may be increased anti-inflammatory activity of the flavonoids in these patients when compared to consuming the intact polyphenol due to increased bioavailability of phase II metabolites as reviewed in Williams et al. [242]. Furthermore, genetic polymorphisms of phase II enzymes may influence the bioavailability of polyphenols as reviewed in Manach et al. [243]. Therefore, in the future, understanding genetic polymorphisms and their effect on phase II polyphenolic metabolites would be useful for reducing variability to polyphenol therapy and improving the efficacy of polyphenols for the management of diabetic complications.

Moreover, given that gut microbiome-transformed metabolites were associated with a beneficial effect on the endothelium long after consumption [239]; it can be recommended that polyphenol benefits in diabetic patients may need to be enhanced with probiotics [244] for the prevention of endothelial damage, which is central to diabetic complications.

In addition, recent evidence suggests that the production of gut microbiome-derived metabolites from polyphenols could differ within individuals. Iglesias-Aguirre et al. [245] observed that resveratrol is metabolized to lunularin by mostly male compared to female individuals. According to Cortés-Martín et al. [246], such variations should be regarded as markers for human polyphenol-related metabotypes. However, given the evidence on the impact of equol metabotype/equol gut microbiota metabolizing ability on cardiometabolic risk factors as reviewed in Frankenfield [247], consideration of human polyphenol-related metabotypes might be important to identify specific metabotypes to maximize the benefits of polyphenols. This is promising for precision medicine. Besides, pharmacological therapy for metabolic syndrome could also have a beneficial influence on prebiotic effect of polyphenols, depending on cardiovascular risk factors and the associated medications [248]. Taken together, further research into individual variations due to sex, race, polyphenol-related metabotypes, and pharmacotherapy presents the opportunity for understanding how favorable drug–polyphenol–gut microbial interactions can be harnessed for the therapy of diabetes and diabetic complications.

Finally, the use of various nanoparticle platforms and encapsulation in food has been shown to improve the efficacy of polyphenols by increasing systemic and local delivery of polyphenols as extensively reviewed in [249,250]. It will be important to consider stability, especially in food particles, and the risk that metal-based and lipid-based nanoparticles may cause oxidative stress. Therefore, the design of polyphenol-based nanoparticles should take into consideration the route and could include a combination of polyphenols to mitigate any potential toxicity due to oxidative stress [251,252]. Further clinical including toxicity studies are required to validate the use of nano-based polyphenols in the management of diabetic complications.

## 7. Conclusions

The development and progression of diabetic complications are associated with various oxidative stress and inflammatory pathways. Various phytochemicals possess antioxidant, anti-inflammatory, and anti-hyperglycemic properties which can potentially prevent or impede the progression of diabetic complications. However, the low bioavailability and absorption of antioxidant phytochemicals currently limit their utility in the clinic. Further studies are needed to further explore the antidiabetic potentials of phytochemicals, improve their absorption and bioavailability, and explore their potential side effects in clinical trials. A future in which more diabetic patients will rely on phytochemicals as an alternative/adjuvant therapy for preventing or treating diabetic complications is one in which bioavailability is improved, and dosage standardization has been achieved either for the crude extracts or the purified bioactive compounds. The hope is that in combination with currently available treatment approaches, antioxidant phytochemicals would significantly reduce the burden of diabetic complications.

## Figures and Tables

**Figure 1 antioxidants-12-00123-f001:**
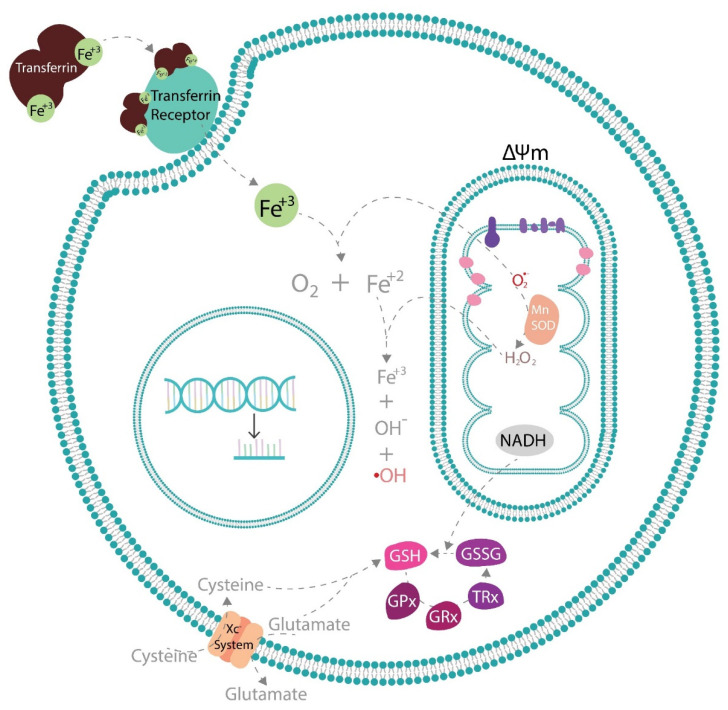
Endogenous antioxidants reactions with reactive oxygen species. MnSOD reacts with superoxide ion (O_2_^−^) converting it to hydrogen peroxide (H_2_O_2_). H_2_O_2_ diffuses to the cytoplasm and is converted to hydroxyl ion (OH^−^) via Fenton reaction with Fe^2+^. OH^−^ cellular damage is prevented by glutaredoxin/thioredoxin, peroxidases (GPx, GRx, TRx), glutathione (GSH), and catalase (not shown). GSH is regenerated by nicotinamide adenine dinucleotide (NADH).

**Table 1 antioxidants-12-00123-t001:** Summary of examples of polyphenols and their effects on endogenous antioxidant systems in diabetic conditions.

Antioxidant System	Examples of Polyphenol	Effects
**UCP2/mitochondrial Complex 1**	Caffeic acid Anthocyanins	In vitro protection of retinal cells, improving mitochondrial function [160]. Improve cardiac function by preserving mitochondrial function in diabetic rat model [166].
**SIRTs**	Resveratrol	Improve GSH, CAT SOD activity and mitochondria function via activation of SIRT1/PGC-1α/FOXO3a.
**GSH/GPx4/Xc^−^ system** **(Inhibition of ferroptosis)**	Glabridin and calycosin	Reno-protective effect in DN rat model [183,184].
	Naringenin	Cardio protective effect [185].
	Astragaloside-IV	In vitro protection of retinal cells [186].
	Quercetin and cryptochlorogenic acid	Preserving β-cell function [87,188].
**Nrf-2**	Epigallocatechin-3-gallate Curcumin	Reno-protective effect in DN [206]. Protective effective against retinal injury in diabetes [213].

## Data Availability

Data is contained within the article.

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
