# Peer review of "Antioxidant Phytochemicals as Potential Therapy for Diabetic Complications"

_antioxidants, 2023, doi:10.3390/antiox12010123_

Round 1

Author Response

We thank the reviewer for a comprehensive feedback on our manuscript. Attached is our response to the reviewers suggestions.

Thank you.

Reviewer 2 Report

The authors provide the detailed analysis and generalization of the contemporary knowledge concerning the effects of polyphenols on endogenous antioxidant systems in living organisms with the emphasis on potential therapeutic strategies for harnessing the relevant properties of polyphenolic compounds to impede the progression of diabetic complications. They conclude that despite the salient benefits of the pertinent phytochemicals their low bioavailability and absorption currently limit their clinical utility. Further studies are needed to explore the antidiabetic potentials of natural compounds, to improve their absorption and bioavailability and to examine their possible side effects in clinical trials. I read this contribution with keen interest and strongly recommend its publication in Antioxidants. In my opinion, the manuscript is well written, and all the material is clearly presented. I have only one suggestion to the Authors. Various physico-chemical/structural parameters such as lipophilicity, hydrogen bonding, molecule size, presence of particular structural features, etc. may be efficiently analyzed in silico for bioantioxidants of interest to estimate the molecular behavior in a living organism and, most prominently, their transport properties, metabolic stability, affinity to proteins and toxicity (e.g., Kancheva V.D. et al. Natural chain-breaking antioxidants and their synthetic analogs as modulators of oxidative stress. Antioxidants 2021, 10, 624). As a possible perspective for research on the optimal choice of the pertinent bioantioxidants, I would suggest mentioning such a possibility.

Author Response

We thank the reviewer for their kind recommendation and Suggestions. Please find in the attached our response to your reviews.

Thank you

Reviewer 3 Report

Dear authors,

After the review process, I have several comments: you should present how was realized the figures in the legends; the paper has a significant lack of information related to the human microbiota; you should add new findings that can explain the correlation between microbiota bioactivity and bioavailability of functional compounds; the altered metabolomic response of human microbiota represents a critical biomarker, that can be used as a clinical target at this moment. 

Best regards!

Author Response

We thank the reviewer for their suggestion. We have included literature as suggested. Thank you

Round 2

Reviewer 3 Report

Dear authors,

the new data related to the correlation between microbiota bioactivity and the bioavailability of functional compounds should be strongly improved

Best regards!

Author Response

We thank the reviewer for their comments. 

We thank the reviewer for his/her summarised assessment of our manuscript and for his/her helpful suggestions to improve our 
manuscript. 1. We have now included information related to the human microbiota including new findings regarding the correlation 
between microbiota bioactivity and bioavailability of functional compounds.

Please refer to Section 6, line 1049-1061.